# The Contribution of Extruded and Fermented Wheat Bran to the Quality Parameters of Wheat Bread, Including the Profile of Volatile Compounds and Their Relationship with Emotions Induced for Consumers

**DOI:** 10.3390/foods10102501

**Published:** 2021-10-18

**Authors:** Elena Bartkiene, Ieva Jomantaite, Ernestas Mockus, Romas Ruibys, Aldona Baltusnikiene, Antonello Santini, Egle Zokaityte

**Affiliations:** 1Department of Food Safety and Quality, Faculty of Veterinary Medicine, Lithuanian University of Health Sciences, Mickeviciaus Str. 9, LT-44307 Kaunas, Lithuania; elena.bartkiene@lsmuni.lt (E.B.); ieva.jomantaite@stud.lsmu.lt (I.J.); 2Institute of Animal Rearing Technologies, Faculty of Animal Science, Lithuanian University of Health Sciences, Mickeviciaus Str. 9, LT-44307 Kaunas, Lithuania; ernestas.mockus@lsmuni.lt; 3Institute of Agricultural and Food Sciences, Agriculture Academy, Vytautas Magnus University, K. Donelaicio Str. 58, LT-44244 Kaunas, Lithuania; romas.ruibys@vdu.lt; 4Department of Biochemistry, Faculty of Medicine, Lithuanian University of Health Sciences, Mickeviciaus Str. 9, LT-44307 Kaunas, Lithuania; aldona.baltusnikiene@lsmuni.lt; 5Department of Pharmacy, University of Napoli Federico II, Via D. Montesano 49, 80131 Napoli, Italy; antonello.santini@unina.it

**Keywords:** wheat bread, wheat bran, extrusion, fermentation, volatile compounds

## Abstract

The aim of this study was to evaluate the contribution of extruded and fermented wheat bran (WBex-f) to the quality of wheat bread (BR), including the volatile compounds (VC) profile and VC relationship with emotions induced for consumers. A comparison study of BR (prepared with 5%, 10%, and 15% untreated wheat bran (nWB) and WBex-f) quality parameters was performed. It was established that nWB increases dough hardness and reduces BR specific volume. The addition of 5% and 10% of WBex-f was not significant on BR porosity and led to the formation of a high number of large pores. nWB and WBex-f increases the mass loss of BR after baking (by 13.38%), and the control breads showed the highest crust darkness, yellowness, and redness. nWB and WBex-f reduces BR firmness during storage, and WBex-f increases the overall acceptability (OA) of BR (by 26.2%). A strong positive correlation was found between OA and the emotion ‘happy’ (r = 0.8696). In BR prepared with WBex-f, a higher content of pyrazine, methyl-; pyrazine, 2-ethyl-; pyrazine, 2-ethyl-6-methyl-; furfural; ethanone, 1-(2-furanyl)-; benzaldehyde; and 3-furanmethanol was found. Finally, it can be stated that WBex-f could prolong the shelf life of BR and leads to the formation of a specific VC profile, which is associated with a higher OA of the product.

## 1. Introduction

Wheat (*Triticum* spp.) is one of the most popular cereal varieties in the world because of its unique technological properties, which are especially suitable for bread preparation. The most popular part of the wheat grain, used for bread making, is the endosperm, as it contains proteins (especially desirable gluten proteins: gliadin and glutenin), which leads to the high porosity of bread. However, despite most of the functional compounds of the wheat grain being located in the outermost tissues [1], these layers of wheat grain have not, until now, been used effectively enough. The use of wheat bran in the food industry is complicated, because the addition of wheat bran to the main food formula leads to products with lower lightness, poorer consistency, harder texture, and lower water binding and gas-holding capacity, as well as lower overall acceptability [2]. In addition, the presence of additional wheat bran in the bread formula decreases bread loaf volume and induces undesirable textural changes and visual modifications compared to refined flour-based bread [2]. For this reason, to increase the effectiveness of wheat bran valorization, pre-treatment technologies to improve its technological, functional, and sensory properties are being studied. Spaggiari et al. [3] reported that fermentation with selected lactic acid bacteria (LAB) strains is a suitable technology for wheat bran valorization, because it is associated with an increase in fermentable substrate bioavailability and the release of bioactive compounds from complex plant matrices, as well as with improved sensory properties of the product [4]. In addition, an extrusion process can be applied for wheat bran pre-treatment, as during this process, there are structural and physicochemical changes to the substrate [5]. Our previous studies showed that the combination of extrusion and fermentation (with selected LAB strains) is a promising technology for wheat bran valorization; as such, pre-treatment enhancement could be applied at industrial scale for the preparation of safer and higher-value food/feed stock [6,7]. The use of extruded bran in whole-wheat bread or bran-enriched breads has been reported [8]. In addition, studies about the effect of adding non-extruded and extruded bran on dough behavior during mixing, forming, and fermentation, as well as on the physical and sensory properties of bread, have been published [9]. Gómez et al. [9] concluded that there are only minimal differences in acceptability between breads made with the different types of bran and, in some cases, the extruded bran showed a clear advantage; however, extruded bran did have different effects on dough rheology, and behavior varied according to the presence or absence of additives. In this study, we hypothesized that the product obtained by using a combination of extrusion and fermentation processes for wheat bran pre-treatment could lead to more effective valorization of wheat bran by including it in the main bread formula; also, as an alternative sourdough, it could improve bread quality parameters as well.

The aim of this study was to evaluate the contribution of extruded/fermented wheat bran to the quality parameters of wheat bread, including the profile of volatile compounds (VC) and their relationship with the emotions induced for consumers. For this purpose, a comparison study of bread (prepared without wheat bran (control) with different quantities (5%, 10%, 15%) of untreated (WB5%, WB10%, WB15%) and extruded/fermented (WB ex-f-5%, WB ex-f-10%, WB ex-f-15%) wheat bran) quality parameters was performed.

## 2. Materials and Methods

### 2.1. Materials Used for Bread Preparation

Wheat flour (type 550D, falling number 350 s, gluten 27%, ash 0.68%) obtained from Kauno Grudai Ltd. mill (Kaunas, Lithuania) was used for the baking of wheat bread. The wheat bread samples were prepared without and with the addition of untreated and extruded fermented wheat bran (5%, 10%, and 15%). Wheat bran samples were obtained from Ustukiu malunas Ltd. mill (Pasvalys, Lithuania). Wheat bran samples extruded at 130 °C at a screw speed of 25 rpm and fermented with a *Lactobacillus uvarum* strain were used for wheat bread preparation (Appendix A. Characteristics of extruded and fermented wheat bran: Appendix A. Acidity, microbiological parameters, and sugar concentration in processed wheat bran. Appendix A. Amino acid concentration (g/100 g) in processed wheat bran. Appendix A. Biogenic amines concentration (mg/kg) in processed wheat bran. Appendix A. Mycotoxin concentration (µg/kg) in processed wheat bran). The pre-treatment technology and characteristics of the untreated and extruded/fermented wheat bran are described in detail by Bartkiene et al. [6].

### 2.2. Bread Preparation

The wheat bread recipe consisted of 1 kg of wheat flour, 1.5% salt, 2% fresh compressed yeast, and 56% water (control bread). Control samples were made without the addition of wheat bran. Different quantities of the extruded/fermented wheat bran were added to the wheat flour (wheat flour 100% plus WB ex-f-5%, WB ex-f-10%, WB ex-f-15%). The water content was calculated as 56% of the total flour and wheat bran mass. The dough was mixed for 2 min at a low-speed regime and then for 10 min at a high-speed regime in a mixer (KitchenAid Artisan, Ohio, USA). After mixing, the dough was left at room temperature (22 ± 2 °C) for 15 min relaxation. After relaxation, the dough was shaped (mass of loaf 400 g), formed, fermented (at 30 ± 2 °C and 80% relative humidity (RH) for 40 min), and baked (in a deck oven (EKA, Borgoricco PD, Italy) at 220 °C for 25 min).

### 2.3. Evaluation of Bread Quality Parameters

After 12 h of cooling at room temperature (22 ± 2 °C), wheat bread samples were subjected to analysis of the following parameters: specific volume, crumb porosity, shape coefficient, mass loss after baking, crust and crumb color, crumb firmness during storage, overall acceptability, emotions induced for consumers by bread, and VC profile.

Bread volume was determined by the rapeseed displacement method [10], and the specific volume was calculated by measuring the volume and weight of the bread loaf and dividing the first by the second. Bread crumb porosity was evaluated by the LST method 1442:(1996) [11]. The bread shape coefficient was calculated by measuring the width and height (measured in mm) of a bread slice and dividing the first by the second.

Mass loss was calculated as a percentage by measuring loaf dough mass before and after baking.

Crust and crumb color parameters were evaluated using a CIE L*a*b* system (CromaMeter CR-400, Konica Minolta, Osaka, Japan).

Dough hardness was measured as the energy required for the sample deformation (CT3 Texture Analyzer, Brookfield, MA, USA). First, 50 g of dough was placed on the Petri dishes and compressed to 10% of its original height at a crosshead speed of 10 mm s^−1^. The resulting peak energy of compression was reported as dough hardness. Five replicates of dough were analyzed and averaged.

Bread crumb hardness during storage in plastic packaging at room temperature (22 ± 2 °C) was determined as the energy required for the sample deformation (CT3 Texture Analyzer, Brookfield, USA). Bread slices of 2 cm thickness were compressed to 10% of their original height at a crosshead speed of 10 mm s^−1^. The resulting peak energy of compression was reported as crumb hardness. Three replicates from three different sets of baking were analyzed and averaged.

Overall acceptability testing of breads was carried out according to ISO method 8586-1 (ISO, 2000) by 20 judges for preliminary sensory acceptability using a 140 mm hedonic line scale ranging from 150 (extremely like) to 0 (extremely dislike).

The breads were also tested (by the same 20 judges) by applying FaceReader 8.0 software (Noldus Information Technology, Wageningen, The Netherlands) (Figure 1), with a scoring scale of eight emotion patterns (neutral, happy, sad, angry, surprised, scared, disgusted, contempt). The whole procedure is described in detail by Bartkiene et al. [12] (Figure 1) (Appendix A. Procedure of the FaceReader analysis). For statistical analysis, the maximum values of the facial expression patterns of the respective sections were used.

The VC of the bread samples were analyzed by gas chromatography-mass spectrometry (GC-MS) as described by Bartkiene et al. [12], with some modifications which are described below. A solid phase microextraction (SPME) device with Stableflex™ fiber coated with a 50 µm PDMS-DVB-Carboxen™ layer (Supelco, St. Louis, MO, USA) was used for analysis. A whole slice of bread was weighed and blended with aqueous sodium chloride solution (30% *w*/*v*) in a ratio of 1 g of bread to 3 mL of NaCl solution. For headspace extraction, 8 g of prepared sample was transferred to the 20 mL extraction vial, sealed with a polytetrafluoroethylene septum, and thermostatted at 60 °C for 15 min before exposing the fiber in the headspace. The fiber was exposed to the headspace of the vial for 10 min and desorbed in an injector liner for 2 min (splitless injection mode). Prepared samples were analyzed with a GCMS-QP2010 (Shimadzu, Japan) gas chromatograph and mass spectrometer. The following conditions were used for analysis: injector temperature 250 °C, ion source temperature 220 °C, interface temperature 260 °C. Helium was used as carrier gas at 0.95 mL min^−1^ flowrate. A Stabilwax-DA capillary column (0.25 mm ID, 0.25 μm film thickness, 30 m length (Restek, USA)) was used for analysis. The temperature gradient was programmed from a start at 40 °C (3 min hold) to 220 °C (6 °C min^−1^) up to 250 °C (10 °C min^−1^) (6 min hold). The VC were identified according to mass spectrum libraries (NIST11, NIST11S, FFNSC2).

### 2.4. Statistical Analysis

The results were expressed as the mean values (for bread samples *n* = 3) ± standard deviation (SD). In order to evaluate the effects of different quantities of wheat bran and different wheat bran pre-treatments on bread quality parameters, data were analyzed by two-way ANOVA and Tukey HSD tests as post hoc tests (statistical program R 3.2.1). The results were recognized as statistically significant at *p* ≤ 0.05.

## 3. Results and Discussion

### 3.1. Influence of Non-Pre-Treated and Extruded/Fermented Wheat Bran on Dough Color Characteristics and Hardness

Bread dough color characteristics and hardness are shown in Table 1. In comparison with the control group dough, the lightness (L*) of the bread doughs prepared with untreated and extruded/fermented wheat bran was, on average, 11.8% and 8.1% lower, respectively. Comparing the redness and/or greenness (a* or −a*, respectively) of the different groups of dough with that of the control group, in all cases, smaller differences were obtained for the doughs prepared with extruded/fermented wheat bran, doughs prepared with 5% extruded/fermented wheat bran showing 21.8 times lower greenness (−a*) than that of the control group. The highest redness (a*) was established for the dough samples prepared with 15% untreated wheat bran; on average, it was 48.2% higher than that of the doughs prepared with the same amount of extruded/fermented wheat bran. The opposite tendency was found for dough yellowness (b*); on average, b* coordinates of doughs prepared with untreated wheat bran were 4.1% lower than those of the control group. By increasing the proportion of extruded/fermented wheat bran in the dough formulation, the b* coordinates of the dough showed a tendency to increase, the highest values being found for the doughs prepared with 15% extruded/fermented wheat bran (22.75 NBS). In comparing dough hardness, different tendencies were established in different sample groups. Comparing control group samples with doughs prepared with untreated wheat bran, hardness was increased by increasing the quantity of untreated wheat bran in the dough formulation. However, samples prepared with 5% untreated wheat bran showed the same hardness as the control group samples (0.2 mJ). Comparing doughs prepared with extruded/fermented wheat bran with those of the control group, no significant differences in hardness were found between groups (for both groups, hardness was, on average, 0.2 mJ).

A first characteristic of wheat bran that should be pointed out is its water absorption, which is different to that of endosperm flour [2]. However, after extrusion, as well as after fermentation, wheat bran characteristics change, and in this study, the hardness of dough prepared with extruded/fermented wheat bran was the same as that of the control dough. In addition, it should be mentioned that in this study, by increasing the wheat bran content in the main bread formula, water content, also, was additionally added. Ugarcic-Hardi, Komlenic, Jukic, Kules, and Jurkin [13] reported that untreated bran had a more negative influence on dough properties in comparison with extruded bran. Pre-treatment of wheat bran using extrusion leads to the solubilization of dietary fiber compounds [14], and extruded bran has a smaller particle size than non-extruded bran, which also has an influence on dough properties [15]. The negative influence on dough quality is attributed to the pre-treated bran disrupting the gluten network more, due to increased flour–bran contact for small particles [15,16]. However, in this study, extruded/fermented wheat bran was used, and such a type of product could be used as an alternative sourdough, because of the high number of viable LAB and low pH, as well as high total titratable acidity values [6]. Finally, no differences in dough hardness were established by including extruded/fermented wheat bran in the main bread formula compared with control doughs.

The results of the ANOVA test indicated that there is a significant effect of adding untreated or extruded/fermented wheat bran (*p* < 0.05) and in different quantities (*p* < 0.05) on dough color characteristics (L*, a*, b*) and dough hardness.

### 3.2. Influence of Non-Pre-Treated and Extruded/Fermented Wheat Bran on Bread Quality Parameters and Changes in Bread Firmness during Storage

The main bread quality characteristics (specific volume, porosity, shape coefficient, mass loss after baking, and color characteristics of the bread crust and crumb) are shown in Table 2. In comparing the specific volume of the different bread groups, it was found that by increasing the quantity of untreated wheat bran in the main bread formula, the specific volume was reduced by 2.2%, 12.1%, and 23.2% compared with the control for bread prepared with 5%, 10%, and 15% bran, respectively. Opposite to these findings, the addition of extruded/fermented wheat bran to the main bread formula did not cause any undesirable changes in specific volume, and in all cases, the specific volume of bread prepared with extruded/fermented wheat bran remained similar to that of the control group breads (on average, 2.78 cm^3^ g^−1^). Similar tendencies were found for bread porosity: increasing the quantity of untreated wheat bran reduced the porosity of the bread: by 10.00% for bread containing 5% untreated wheat bran, by 17.7% for bread containing 10% untreated wheat bran, and by 32.1% for bread containing 15% untreated wheat bran. However, the addition of 5% and 10% extruded/fermented wheat bran had no significant effect on bread porosity, but 15% extruded/fermented wheat bran reduced bread porosity by 4.0% in comparison with control breads. In addition, as can be seen from the images of the bread texture (Table 2), the incorporation of extruded/fermented wheat bran in the main bread formula led to the formation of a high number of large pores compared with control breads, as well as in the breads prepared with untreated wheat bran. The results of the ANOVA test indicated that there is a significant effect of adding untreated or extruded/fermented wheat bran and of the different quantities, and the interaction of these factors on bread specific volume (*p* < 0.0001) and porosity (*p* < 0.0001). In comparing bread shape coefficients between the control group and breads prepared with extruded/fermented wheat bran, no significant differences were found; however, the shape coefficient of bread prepared with 5% untreated wheat bran was 7.7% higher, and that of bread prepared with 15% untreated wheat bran was 20.6% lower than that of the control bread group. In all cases, the addition of wheat bran (untreated and extruded/fermented) increased mass loss after baking: in the group prepared with untreated wheat bran, mass loss after baking was, on average, 11.39% higher, and in the group prepared with extruded/fermented wheat bran, it was, on average, 15.36% higher than that of the control group; these results were not significantly influenced by the quantity of untreated or treated wheat bran used.

In comparing bread crust color coordinates, the highest L* was shown by control breads, as well as by breads prepared with 10% extruded/fermented wheat bran (on average, 52.95 NBS). The highest crust a* was found for control group samples; a* values were 13.0% and 32.4% lower for breads prepared with untreated and extruded/fermented wheat bran, respectively. In addition, all the breads prepared with untreated and extruded/fermented wheat bran showed lower crust b* in comparison with the control group (on average, 18.7% and 17.2% lower, respectively).

Comparing bread crumb color coordinates, the highest L* and b* coordinates were shown by control breads (80.14 and 24.13 NBS, respectively), and increasing the quantity of wheat bran (untreated and extruded/fermented) in the main bread formula reduced the L* coordinates of the bread crumb. Comparing bread crumb a* coordinates, the lowest a* was found for the control bread (−1.14 NBS) and, in all cases, increasing the content of wheat bran (untreated and extruded/fermented) in the bread formula increased the a* of the bread crumb.

According to Wang et al. [8], supplementation of bread with extruded bran reduces volume in comparison with bread prepared with untreated bran. However, these changes are related to bran extrusion parameters, and the volume of breads prepared with bran extruded at medium or high speed was similar to that of breads prepared with untreated bran [2]. Gómez et al. [9] reported that breads prepared with extruded bran and an improver have a higher volume than those prepared with untreated bran and an improver; these findings were explained by the amylolytic hydrolysis of modified starch in extruded bran by the improver used, as these changes led to higher gas production because of a higher release of fermentable sugars from bran. After extrusion, the water absorption of wheat bran increases in comparison with that of untreated wheat bran [8,14], and these changes could be associated with changes in the bread structure. In addition, as a separate treatment of bran, fermentation has been described. Katina et al. [17] reported that the addition of 20 h yeast-fermented pearled kernels to bread formulation increases bread specific volume and leads to a softer crumb compared to the addition of untreated bran. These changes were explained by possible arabinoxylan solubilization due to the fermentation. Coda et al. [18] reported that wheat bran fermented for 8 h with a combination of *Lactobacillus brevis* and yeast strains increases bread volume, and this is related to better dough stability as well as increased gas retention. In addition, Gobbetti, Corsetti, and Rossi [19] published that fermentation by yeast and heterofermentative LAB is more effective, and fermented bran can improve bread quality due to the increased acidification parameters of the dough [2]. Opposite to the incorporation of fermented bran, the inclusion of unfermented bran in the main bread formula can disturb the gas cell wall structure of the dough [20] and lead to a low bread volume and dense crumb texture. Finally, there are no published data about the incorporation of extruded/fermented wheat bran in the main bread formula; however, the results obtained in this study are prospective for the preparation of higher-value wheat bread at industrial scale.

Changes of the bread firmness during storage in plastic packaging at room temperature are shown in Figure 2. After 12 h of storage, the control breads and breads prepared with the addition of 10% untreated wheat bran showed the same hardness values (0.2 mJ); bread of other groups was harder: deformation of 0.25 mJ was determined for all of them except for bread prepared with 15% extruded/fermented wheat bran (0.30 mJ). After 24 h of storage, the smallest changes in hardness were found in the breads prepared with 10% untreated wheat bran (0.25 mJ), and the hardest breads were those prepared with 10% extruded/fermented wheat bran (0.80 mJ). After 48 h of storage, the texture tendencies of breads prepared with 10% extruded/fermented wheat bran remained the same, as they had the hardest texture (0.85 mJ). After 72, 96, and 120 h of storage, tendencies remained the same: the control breads had the hardest texture (1.40, 1.90, and 2.55 mJ, respectively), and the softest breads were those prepared with 15% untreated wheat bran (0.45, 0.50, and 0.65 mJ, respectively). Finally, comparing all the sample groups with the control group, after 120 h of storage, all groups showed lower hardness values (54.9% lower for bread prepared with 5% untreated wheat bran, 43.1% lower for bread prepared with 10% untreated wheat bran, and 74.5% lower for bread prepared with 15% untreated wheat bran; 41.2% lower for bread prepared with 5% extruded/fermented wheat bran, 9.8% lower for bread prepared with 10% extruded/fermented wheat bran, and 27.5% lower for bread prepared with 15% extruded/fermented wheat bran). According to Katina et al. [21], a combination of bran fermentation with enzymatic treatment not only improves the loaf volume and textural properties of bread but also prolongs the shelf life [22], and these effects are mainly due to the redistribution of water among starch, gluten, and bran particles during storage [21]. Our results are in agreement with these findings as well as with those of Hemdane et al. [23], who reported that the incorporation of bran in the main bread formula leads to a greater stability of bread texture during storage, and this could be related to the retardation of amylopectin retrogradation in the presence of bran. The results of the ANOVA test indicated that there is a significant effect of adding untreated or extruded/fermented wheat bran (*p* < 0.0001) and in different quantities (*p* < 0.0001), and the interaction of these factors (*p* < 0.0001) on bread hardness after 120 h of storage.

### 3.3. Overall Acceptability and Emotions Induced for Consumers of Bread Prepared with Different Quantities of Untreated and Extruded/Fermented Wheat Bran

The overall acceptability and emotions induced for consumers by bread prepared with different quantities of untreated and extruded/fermented wheat bran are shown in Table 3. The lowest overall acceptability was shown for control bread and bread prepared with untreated wheat bran (on average, 97.0 mm). All the breads prepared with extruded/fermented wheat bran showed, on average, 26.2% higher overall acceptability compared with the control as well as with bread prepared with untreated wheat bran. A strong positive correlation was found between the overall acceptability of the breads and the emotion ‘happy’ (r = 0.8696). In addition, moderate negative correlations were found between the overall acceptability of the bread and the emotions ‘sad’ and ‘contempt’ (r = −0.5120 and r = −0.6399, respectively). A positive moderate correlation was also established between the breads’ overall acceptability and the emotion ‘scared’ (r = 0.5535). There are reports about the negative influence of adding wheat bran to the main bread formula on bread quality, including sensory characteristics [23,24]. It has been reported that the sensory properties of bread containing extruded bran are worse than those of control breads but still acceptable [2]. According to Gómez et al. [9], the sensory properties of breads prepared with untreated and extruded bran are similar.

The results of the ANOVA test indicated that there is a significant effect of adding different types (untreated or extruded/fermented) of wheat bran on breads’ overall acceptability (*p* < 0.0001) and the induced emotion ‘happy’ (*p* < 0.05).

Usually, traditional sensory analysis tests are not able to sufficiently predict market performance because most of these test techniques are based on self-reports and, therefore, underlie a conscious/rational decision-making process [23,25]. Consequently, the long-term consumer acceptance of special foods is not adequately reflected by using traditional methods [25]. In addition, when consumers make a purchase, they tend to make their decision based on emotions and justify the decision with logic later on. From this point of view, studying the emotions induced by novel products and/or recipes becomes very important, as it could lead to better understanding of the success of new technological solutions. This study showed that there is a strong positive correlation between the overall acceptability of bread and the emotion ‘happy’, and, by reducing bread overall acceptability, expression of the emotions ‘sad’ and ‘contempt’ is increased.

### 3.4. Volatile Compound Profile of Bread Prepared with Different Quantities of Untreated and Extruded/Fermented Wheat Bran

The VC found in the breads prepared with different quantities of untreated and extruded/fermented wheat bran are shown in Table 4. Considering all bread samples, the predominant VC were ethanol; 1-butanol, 3-methyl-; 1-hexanol; furfural; 3-furanmethanol; and phenethyl alcohol, and the lowest content of ethanol; 1-butanol, 3-methyl-; pyrazine, 2-ethyl-6-methyl-; furfural; and 2-furancarboxaldehyde, 5-methyl- was established in control bread group samples (except for 2-furancarboxaldehyde, 5-methyl-, the content of which was similar in control samples to that in bread prepared with 5% untreated wheat bran). Comparing breads prepared with untreated and extruded/fermented wheat bran, a higher content of pyrazine, methyl-; pyrazine, 2-ethyl-; pyrazine, 2-ethyl-6-methyl-; furfural; ethanone, 1-(2-furanyl)-; benzaldehyde; and 3-furanmethanol was found in all breads prepared with extruded/fermented wheat bran. However, a higher content of 1-butanol, 3-methyl-; 1-hexanol; and 3-nonen-1-ol, (Z) was established in breads prepared with untreated wheat bran than in breads prepared with extruded/fermented wheat bran. Strong and very strong positive correlations were found between the overall acceptability of bread samples and the VC: pyrazine, 2-ethyl-6-methyl-; pyrazine, 2-ethyl-3-methyl-; furfural; ethanone, 1-(2-furanyl)-; benzaldehyde; 2-furancarboxaldehyde, 5-methyl-; and maltol (Table 4). Meanwhile, strong and very strong negative correlations were established between the overall acceptability of bread samples and the VC: 1-hexanol; formic acid, heptyl ester; 2-propanol, 1-(2-methoxy-1-methylethoxy)-; 1-octanol; 3-nonen-1-ol, (Z)-; propanoic acid, 2-methyl-, 3-hydroxy-2,4,4-trimethylpentyl ester; propanoic acid, 2-methyl-, 2,2-dimethyl-1-(2-hydroxy-1-methylethyl) propyl ester; 2(3H)-furanone, dihydro-5-pentyl-; and indole. Similar tendencies were established with the emotion ‘happy’ and VC content, and very strong and strong positive correlations were found between the emotion ‘happy’ and the VC: pyrazine, methyl-; pyrazine, 2-ethyl-6-methyl-; pyrazine, 2-ethyl-3-methyl-; furfural; ethanone, 1-(2-furanyl)-; benzaldehyde; 3-furanmethanol; and maltol. Strong and very strong negative correlations were established between the emotion ‘happy’ and the VC: 1-hexanol; nonanal; formic acid, heptyl ester; 2-propanol, 1-(2-methoxy-1-methylethoxy)-; 1-octanol; butanoic acid; 3-nonen-1-ol, (Z)-; propanoic acid, 2-methyl-, 3-hydroxy-2,4,4-trimethylpentyl ester; propanoic acid, 2-methyl-, 2,2-dimethyl-1-(2-hydroxy-1-methylethyl) propyl ester; 2(3H)-furanone, dihydro-5-pentyl-; p-cresol; indole; and 1H-indole, 3-methyl-.

Pyrazines are heterocyclic compounds responsible for a ‘roasted-like’ flavor in food, and furfural possesses a sweet, woody, almond, baked bread odor, as well as a sweet, woody, bready, nutty, caramel-like taste with a burnt astringent nuance. The ethanone, 1-(2-furanyl)- odor type is described as high-strength balsamic (sweet, balsamic, almond, cocoa, caramel-like, coffee). A negative correlation was also found between overall acceptability and 1-hexanol, the odor of which is described as pungent, etherial, fusel oil, fruity, alcoholic, and sweet with a green top note. In addition, a negative correlation was found between overall acceptability and formic acid, which is one of the main VC in black tea and in acid-hydrolyzed soy sauce. A negative influence on overall acceptability was shown by 1-octanol, the odor of which is described as waxy, green, citrus, aldehydic, and floral with a sweet, fatty, coconut nuance, as well as by 3-nonen-1-ol, (Z)-, the odor of which is associated with fresh, waxy, green melon, rind, and tropical mushroom, and by propanoic acid, 2-methyl-, 2,2-dimethyl-1-(2-hydroxy-1-methylethyl) propyl ester, the odor of which is described as etherial, diffusive, fruity, sweet, and tutti-frutti. In addition, 2(3H)-furanone, dihydro-5-pentyl-, the odor of which is described as sweet, coconut, coumarin, lactonic, creamy, and powdery, was associated with lower overall acceptability of the breads. Indole (whose odor is characterized as pungent, floral, slightly naphtha- and mothball-like with a fecal and animalic musty character) also showed a negative correlation with the overall acceptability of the bread. Finally, it can be stated that extruded/fermented wheat bran leads to the formation of a specific bread odor, which is associated with the higher overall acceptability of the product.

## 4. Conclusions

Finally, untreated wheat bran increases dough hardness and reduces the specific volume of bread, and the addition of 5% and 10% extruded and fermented wheat bran does not have a significant effect on bread porosity, leading to the formation of a high number of large pores. The inclusion of both untreated and extruded/fermented wheat bran increases the mass loss of bread after baking (by 13.38%) and reduces crust L*, a*, and b* coordinates in comparison with control bread. In addition, untreated and extruded/fermented wheat bran reduces bread firmness during storage, and extruded/fermented wheat bran increases the overall acceptability of bread (by 26.2%). In addition, a strong positive correlation was found between the overall acceptability of bread and the emotion ‘happy’ (r = 0.8696). In bread prepared with extruded/fermented wheat bran, a higher content of pyrazine, methyl-; pyrazine, 2-ethyl-; pyrazine, 2-ethyl-6-methyl-; furfural; ethanone, 1-(2-furanyl)-; benzaldehyde; and 3-furanmethanol was found. According to the results obtained, it can be stated that extruded/fermented wheat bran could prolong the shelf life of bread and leads to the formation of a specific VC profile that is associated with the higher overall acceptability of the product.

## Figures and Tables

**Figure 1 foods-10-02501-f001:**
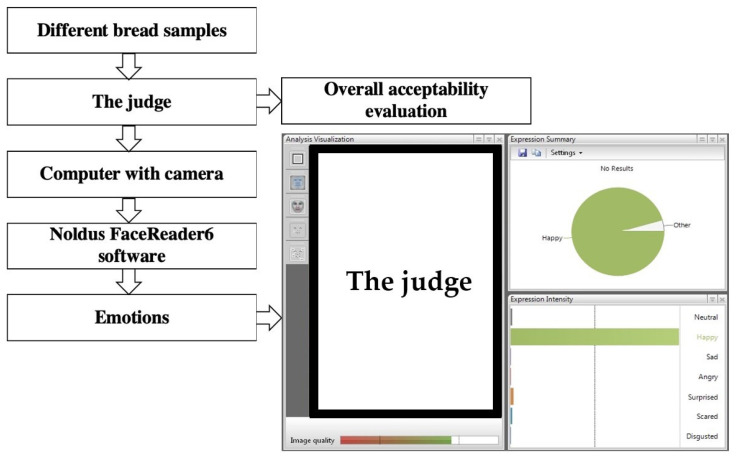
Analysis of the emotions induced by the bread using FaceReader 8 software (Noldus Information Technology, Wageningen, The Netherlands) and further scoring the eight emotion patterns: neutral, happy, sad, angry, surprised, scared, disgusted, and contempt.

**Figure 2 foods-10-02501-f002:**
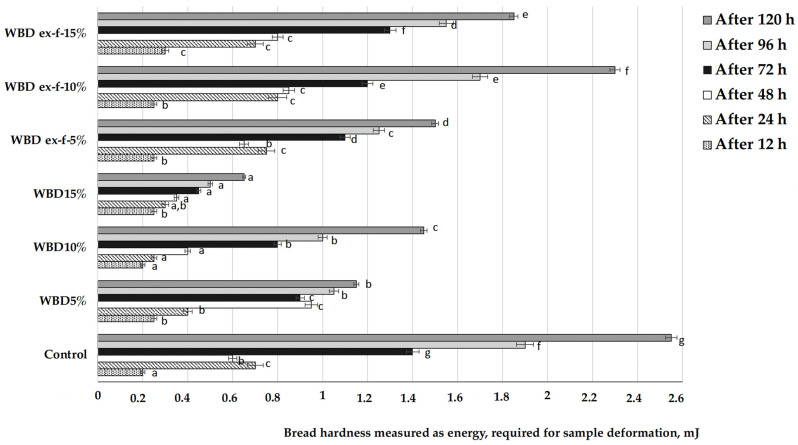
Changes of bread firmness during storage in plastic packaging at room temperature (data expressed as mean values (*n* = 3) ± standard deviation (SD); a–g Mean values within a column with different letters are significantly different (*p* ≤ 0.05)). The control bread was produced without wheat bran; WB5%, WB10%, WB15%—bread produced with 5%, 10%, and 15%, respectively, untreated wheat bran; WB ex-f-5%, WB ex-f-10%, WB ex-f-15%—bread produced with 5%, 10%, and 15%, respectively, extruded fermented wheat bran.

**Table 1 foods-10-02501-t001:** Dough color characteristics and hardness.

Bread Samples	Color Characteristics			Dough Hardness, mJ
	L*	a*	b*	
Control	95.1 ± 0.97 e	−1.31 ± 0.05 a	21.49 ± 0.31 b	0.2 a
	Dough with untreated wheat bran			
WB5%	88.3 ± 0.13 c	1.14 ± 0.20 d	20.81 ± 0.43 a	0.2 a
WB10%	84.3 ± 1.28 a	2.55 ± 0.05 f	20.20 ± 0.41 a	0.3 b
WB15%	79.1 ± 0.87 b	3.63 ± 0.36 g	20.80 ± 0.38 a	0.4 c
	Dough with extruded fermented wheat bran			
WB ex-f-5%	91.5 ± 0.60 d	−0.06 ± 0.02 b	21.90 ± 0.29 b	0.2 a
WB ex-f-10%	87.9 ± 1.97 c	0.79 ± 0.12 c	22.30 ± 0.42 b	0.2 a
WB ex-f-15%	82.9 ± 1.70 a	1.88 ± 0.06 e	22.75 ± 0.23 b,c	0.2 a

Data expressed as mean values (*n* = 5) ± standard deviation (SD). a–e. Mean values within a row with different letters are significantly different (*p* ≤ 0.05). L* lightness; a* redness or −a* greenness; b* yellowness or −b* blueness; NBS—National Bureau of Standards units. The control dough was produced without wheat bran; WB5%, WB10%, WB15%—dough produced with 5%, 10%, 15%, respectively, untreated wheat bran; WB ex-f-5%, WB ex-f-10%, WB ex-f-15%—bread produced with 5%, 10%, 15%, respectively, extruded fermented wheat bran.

**Table 2 foods-10-02501-t002:** Bread specific volume, porosity, shape coefficient, mass loss after baking, color characteristics of the bread crust and crumb, and bread crumb images.

Bread Samples	Specific Volume, cm^3^ g^−1^	Porosity, %		Shape Coefficient	Mass Loss after Baking, %	
Control	2.72 ± 0.04 d	68.3 ± 0.98 e		2.48 ± 0.14 b	8.83 ± 2.3 a	
	Wheat bread with untreated wheat bran					
WB5%	2.66 ± 0.02 c	61.5 ± 1.2 c		2.67 ± 0.15 c	12.08 ± 1.28 b	
WB10%	2.39 ± 0.07 b	56.5 ± 2.5 b		2.40 ± 0.18 b	11.75 ± 0.90 b	
WB15%	2.09 ± 0.03 a	46.4 ± 1.1 a		1.97 ± 0.07 a	10.33 ± 0.52 b	
	Wheat bread with extruded fermented wheat bran					
WB ex-f-5%	2.78 ± 0.03 d	68.5 ± 0.9 e		2.48 ± 0.12 b	15.00 ± 4.82 c	
WB ex-f-10%	2.80 ± 0.04 d	69.3 ± 0.6 e		2.48 ± 0.17 b	16.33 ± 1.15 c,d	
WB ex-f-15%	2.76 ± 0.02 d	65.6 ± 1.6 d		2.43 ± 0.19 b	14.75 ± 1.32 c	
Bread samples	Color characteristics					
	Crust			Crumb		
	L*	a*	b*	L*	a*	b*
Control	53.14 ± 1.55 d	12.97 ± 1.03 e	26.44 ± 0.87 c	80.14 ± 0.23 d	−1.14 ± 0.09 a	24.13 ± 0.21 e
	Wheat bread with untreated wheat bran					
WB5%	48.84 ± 2.90 b	11.57 ± 0.21 d	22.55 ± 2.27 a,b	71.14 ± 0.02 c	2.21 ± 0.07 d	22.18 ± 0.21 c
WB10%	43.33 ± 1.03 a	11.73 ± 0.85 d	20.33 ± 1.90 a	67.96 ± 0.84 b	3.39 ± 0.24 e	20.69 ± 0.11 a
WB15%	45.90 ± 2.99 a,b	10.58 ± 0.24 c	21.62 ± 2.36 a	62.07 ± 1.24 a	4.95 ± 0.37 f	20.93 ± 0.30 a
	Wheat bread with extruded fermented wheat bran					
WB ex-f-5%	50.08 ± 3.95 b,c	9.33 ± 0.49 b	22.55 ± 1.36 a,b	70.86 ± 0.92 c	0.64 ± 0.13 b	22.81 ± 0.12 d
WB ex-f-10%	52.75 ± 1.52 d	8.72 ± 0.40 a	23.29 ± 1.00 b	66.24 ± 1.51 b	1.69 ± 0.04 c	22.23 ± 0.09 c
WB ex-f-15%	48.25 ± 2.50 b	8.25 ± 1.35 a	19.85 ± 0.94 a	62.84 ± 1.20 a	2.34 ± 0.09 d	21.40 ± 0.32 b
Bread crumb images						
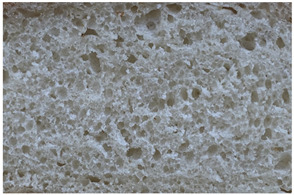						
Control bread						
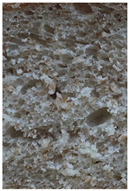		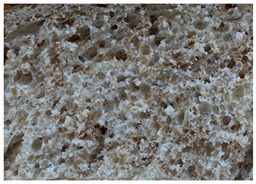			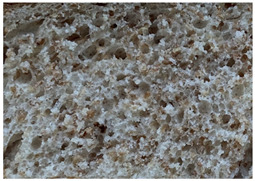	
WB5%		WB10%			WB15%	
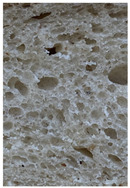		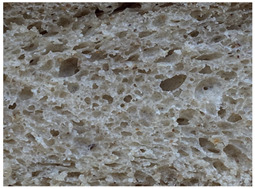			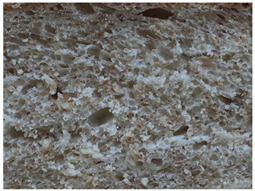	
WB ex-f-5%		WB ex-f-10%			WB ex-f-15%	

Data expressed as mean values (*n* = 5) ± standard deviation (SD). a–f. Mean values within a row with different letters are significantly different (*p* ≤ 0.05). L* lightness; a* redness or −a* greenness; b* yellowness or −b* blueness; NBS—National Bureau of Standards units. The control bread was produced without wheat bran; WB5%, WB10%, WB15%—bread produced with 5%, 10%, 15%, respectively, untreated wheat bran; WB ex-f-5%, WB ex-f-10%, WB ex-f-15%—bread produced with 5%, 10%, 15%, respectively, extruded fermented wheat bran.

**Table 3 foods-10-02501-t003:** Overall acceptability and emotions induced for consumers by breads enriched with untreated and extruded fermented wheat bran.

Bread Samples	Overall Acceptability	Emotions Induced by the Bread (From 0 to 1)								
		Neutral	Happy	Sad	Angry	Surprised	Scared	Disgusted	Contempt	Valence
Control	88.2 ± 10.1 a	0.8431 ± 0.0045 d	0.0248 ± 0.0019 a	0.0056 ± 0.0004 b	0.0117 ± 0.0011 b	0.0005 ± 0.0002 a	0.0002 ± 0.0001 a	0.0004 ± 0.0001 a	0.0032 ± 0.0003 b	0.0576 ± 0.0042 f
WB5%	105.3 ± 12.4 a	0.8916 ± 0.0031 f	0.0390 ± 0.0016 c	0.0241 ± 0.0021 c	0.0129 ± 0.0014 b	0.0102 ± 0.0009 c	0.0002 ± 0.0001 a	0.0012 ± 0.0001 c	0.0003 ± 0.0001 a	0.0006 ± 0.0001 a
WB10%	99.4 ± 8.6 a	0.6980 ± 0.0063 a	0.0270 ± 0.0019 b	0.0203 ± 0.0014 c	0.0730 ± 0.0023 d	0.0020 ± 0.0001 b	0.0002 ± 0.0001 a	0.0008 ± 0.0001 b	0.0050 ± 0.0004 c	0.0430 ± 0.0036 e
WB15%	95.2 ± 9.7 a	0.8270 ± 0.0059 c	0.0360 ± 0.0032 c	0.0209 ± 0.0017 c	0.0150 ± 0.0012 c	0.0024 ± 0.0002 b	0.0001 ± 0.0001 a	0.0012 ± 0.0001 c	0.0068 ± 0.0005 d	0.0500 ± 0.0049 f
WB ex-f-5%	131.6 ± 8.4 a	0.9200 ± 0.0103 g	0.0470 ± 0.0038 d	0.0040 ± 0.0002 b	0.0006 ± 0.0001 a	0.0009 ± 0.0002 a	0.0007 ± 0.0002 b	0.0011 ± 0.0002 c	0.0004 ± 0.0001 a	0.0040 ± 0.0003 c
WB ex-f-10%	130.4 ± 9.4 a	0.8610 ± 0.0114 e	0.0451 ± 0.0034 d	0.0030 ± 0.0004 a	0.0013 ± 0.0002 a	0.0022 ± 0.0001 b	0.0001 ± 0.0001 a	0.0008 ± 0.0001 b	0.0003 ± 0.0001 a	0.0030 ± 0.0002 b
WB ex-f-15%	132.9 ± 6.9 a	0.7525 ± 0.0127 b	0.0472 ± 0.0039 d	0.0031 ± 0.0001 a	0.0008 ± 0.0001 a	0.0014 ± 0.0003 a	0.0028 ± 0.0003 c	0.0004 ± 0.0001 a	0.0004 ± 0.0001 a	0.0137 ± 0.0021 d

Data expressed as mean values (*n* = 20) ± standard deviation (SD). a–g. Mean values within a row with different letters are significantly different (*p* ≤ 0.05). The control bread was produced without wheat bran; WB5%, WB10%, WB15%—bread produced with 5%, 10%, and 15%, respectively, untreated wheat bran; WB ex-f-5%, WB ex-f-10%, WB ex-f-15%—bread produced with 5%, 10%, and 15%, respectively, extruded fermented wheat bran.

**Table 4 foods-10-02501-t004:** Volatile compound profile of breads enriched with untreated and extruded and fermented wheat bran.

RT, min	Volatile Compounds	Bread Samples							r	
		Control	WB5%	WB10%	WB15%	WB ex-f-5%	WB ex-f-10%	WB ex-f-15%	OA/VC	‘H’/VC
4.066	Ethanol	9.91 ± 0.32 a	20.2 ± 1.3 e	16.3 ± 0.4 d	15.0 ± 0.5 c	13.1 ± 0.9 b	15.9 ± 0.3 d	12.9 ± 1.1 b	0.0402	0.1578
10.336	1-Butanol, 3-methyl-	8.63 ± 0.41 a	14.0 ± 0.2 e	15.4 ± 0.5 f	12.3 ± 0.1 c	12.4 ± 0.2 c	13.3 ± 0.4 d	11.8 ± 0.2 b	0.2410	0.1523
12.011	Pyrazine, methyl-	3.72 ± 0.21 c	3.21 ± 0.17 b	2.48 ± 0.19 a	3.74 ± 0.23 c	4.04 ± 0.12 d	4.29 ± 0.17 d	4.20 ± 0.15 d	0.4876	0.6595
13.546	Pyrazine, 2,6-dimethyl-	1.72 ± 0.09 b	1.02 ± 0.21 a	1.35 ± 0.18 a	1.51 ± 0.14 a,b	2.03 ± 0.32 b,c	1.56 ± 0.11 a,b	1.84 ± 0.13 b	0.3971	0.3338
13.88	Pyrazine, 2-ethyl-	3.63 ± 0.25 c,d	2.49 ± 0.19 a	2.77 ± 0.23 a,b	3.51 ± 0.14 c	4.08 ± 0.28 d	3.39 ± 0.17 c	3.43 ± 0.16 c	0.2507	0.2877
14.004	1-Hexanol	12.4 ± 0.1 e	11.8 ± 0.2 d	11.6 ± 0.3 d	10.8 ± 0.10 c	9.00 ± 0.11 b	8.91 ± 0.07 b	8.71 ± 0.05 a	−0.8314	−0.8516
14.884	Pyrazine, 2-ethyl-6-methyl-	0.78 ± 0.05 a	0.87 ± 0.04 a	1.03 ± 0.05 b	1.13 ± 0.08 b	1.66 ± 0.18 d	1.39 ± 0.11 c	1.65 ± 0.21 d	0.8159	0.8071
15.062	Nonanal	3.74 ± 0.08 d	3.55 ± 0.09 c	6.03 ± 0.25 e	2.40 ± 0.13 b	2.07 ± 0.09 a	2.47 ± 0.18 b	2.44 ± 0.15 b	−0.4331	−0.7344
15.39	Pyrazine, 2-ethyl-3-methyl-	1.70 ± 0.08 b	1.46 0.05 a	1.60 ± 0.04 b	1.79 ± 0.12 b,c	2.46 ± 0.23 d	1.82 ± 0.05 c	2.07 ± 0.11 c,d	0.6028	0.6020
16.54	Formic acid, heptyl ester	1.07 ± 0.04 f	0.86 ± 0.04 d	0.96 ± 0.02 e	0.70 ± 0.05 c	0.29 ± 0.03 a	0.49 ± 0.05 b	0.69 ± 0.03 c	−0.7211	−0.8340
16.894	Furfural	7.87 ± 0.12 a	8.39 ± 0.23 b	10.1 ± 1.4 b	13.3 ± 0.9 c	16.0 ± 0.7 e	14.8 ± 0.3 d	14.7 ± 0.4 d	0.7379	0.7770
17.116	2-Propanol, 1-(2-methoxy-1-methylethoxy)-	1.65 ± 0.18 d	1.07 ± 0.04 c	1.12 ± 0.03 c	0.97 ± 0.03 b	0.91 ± 0.02 b	1.15 ± 0.04 c	0.78 ± 0.09 a	−0.6016	−0.7147
17.836	Ethanone, 1-(2-furanyl)-	1.42 ± 0.04 a	1.47 ± 0.08 a	1.97 ± 0.07 b	2.07 ± 0.10 b,c	2.54 ± 0.11 d	2.52 ± 0.14 d	2.56 ± 0.12 d	0.7890	0.7375
18.043	2-Propanol, 1-(2-methoxypropoxy)-	2.29 ± 0.14 e	1.48 ± 0.06 b	1.74 ± 0.10 c	1.73 ± 0.09 c	1.51 ± 0.07 b	1.96 ± 0.08 d	1.15 ± 0.09 a	−0.4480	−0.5199
18.31	Benzaldehyde	1.57 ± 0.11 a	1.71 ± 0.04 a,b	1.49 ± 0.09 a	1.42 ± 0.07 a	1.97 ± 0.02 c	1.88 ± 0.06 b	2.59 ± 0.14 d	0.7087	0.6976
18.411	2-Nonenal, (E)-	3.41 ± 0.21 e	1.82 ± 0.03 c	1.79 ± 0.04 b,c	1.69 ± 0.03 b	1.75 ± 0.06 b	1.56 ± 0.02 a	2.38 ± 0.11 d	−0.3672	−0.4587
18.736	1-Octanol	0.750.06 e	0.57 ± 0.03 c,d	0.59 ± 0.02 d	0.49 ± 0.03 c	0.24 ± 0.02 a	0.34 ± 0.03 b	0.42 ± 0.02 c	−0.8204	−0.8799
19.387	2-Furancarboxaldehyde, 5-methyl-	1.50 ± 0.04 a	1.56 ± 0.03 a	2.26 ± 0.12 b	3.09 ± 0.15 c	4.38 ± 0.10 e	4.08 ± 0.11 d	4.99 ± 0.17 f	0.7952	0.7942
19.515	Hexadecane	0.74 ± 0.06 c	0.58 ± 0.04 b	0.37 ± 0.02 a	0.43 a,b	0.39 ± 0.03 a	0.55 ± 0.04 b	0.39 ± 0.02 a	−0.4945	−0.4305
20.35	Butanoic acid	0.29	nd	nd	nd	nd	nd	nd	−0.5373	−0.6070
21.04	3-Furanmethanol	8.78 ± 0.15 c,d	8.64 ± 0.12 c	8.04 ± 0.10 a	8.35 ± 0.09 b	11.42 ± 0.32 f	8.89 ± 0.10 d	9.53 ± 0.18 e	0.5850	0.6154
21.438	3-Nonen-1-ol, (Z)-	0.58 ± 0.02 c	0.57± 0.04 c	0.63 ± 0.03 d	0.60 ± 0.04 d	0.25 ± 0.03 a	0.27 ± 0.04 a	0.37 ± 0.03 b	−0.8235	−0.8100
22.706	D-Carvone	nd	0.28 0.01 a	nd	0.34 ± 0.03 b	nd	nd	nd	−0.4434	−0.0528
23.554	3-Hydroxypyridine monoacetate	nd	nd	nd	nd	nd	nd	0.85 ± 0.09	0.4405	0.4231
23.725	Dec-(4Z)-en-1-ol	0.35 ± 0.03 b	0.28 ± 0.02 a	0.80 ± 0.07 e	0.45 ± 0.04c	nd	nd	0.55 ± 0.03 d	−0.3827	−0.5316
24.175	2,4-Decadienal, (E,E)-	0.66 ± 0.03d	0.36 ± 0.02 a	0.93 ± 0.07 e	0.54 ± 0.04 c	0.43 ± 0.02 b	0.44 0.03 b	0.82 ± 0.06 e	−0.1232	−0.4300
24.48	1H-Pyrrole, 1-(2-furanylmethyl)-	nd	nd	nd	0.23 ± 0.03a	0.33b	nd	0.33± 0.05b	0.4272	0.5704
24.64	Hexanoic acid	1.24 ± 0.11 b	1.50 ± 0.14 c	1.34 ± 0.18 b,c	2.94 ± 0.23 e	1.02 ± 0.14 a	1.57 ± 0.11 c	1.97 ± 0.15 d	−0.2091	0.0734
24.871	5,9-Undecadien-2-one, 6,10-dimethyl-, (Z)-	1.27 ± 0.02 b	nd	nd	0.22 ± 0.01 a	0.18 ± 0.03 a	0.24 ± 0.02 a	nd	−0.5021	−0.5366
25.101	Propanoic acid, 2-methyl-, 3-hydroxy-2,4,4-trimethylpentyl ester	1.95 ± 0.14 d	1.43 ± 0.17 c	0.90 ± 0.07 b	1.01 ± 0.08 b	0.50 ± 0.04 a	0.97 ± 0.07 b	0.52± 0.03 a	−0.7329	−0.6900
25.54	Propanoic acid, 2-methyl-, 2,2-dimethyl-1-(2-hydroxy-1-methylethyl)propyl ester	1.72 ± 0.19 e	1.44 ± 0.21 d	0.97 ± 0.18 b,c	0.99 ± 0.14 b,c	0.47 ± 0.03 a	0.78 ± 0.05 b	0.53 ± 0.04 a	−0.7835	−0.7364
26.051	Phenethyl alcohol	5.95 ± 0.41 e	4.82 ± 0.34 d	3.16 ± 0.11 c	2.43 ± 0.19 a	2.69 ± 0.18 b	2.58 ± 0.21 a,b	2.53 ± 0.14 a	−0.5360	−0.4593
27.134	Maltol	nd	nd	nd	1.54 ± 0.13 a	1.88 ± 0.9 b	1.82 ± 0.07 b	1.96 ± 0.11 b,c	0.6769	0.7943
28.279	2(3H)-Furanone, dihydro-5-pentyl-	0.56 ± 0.02 b	0.85 ± 0.04 c	1.13 0.05 d	1.27 ± 0.09 d,e	nd	nd	0.34 ± 0.03 a	−0.6956	−0.6229
29.072	p-Cresol	3.30 ± 0.22 b	0.65 ± 0.05 a	nd	nd	nd	nd	nd	−0.5728	−0.6052
31.07	Guaiacol, 4-vinyl-	nd	nd	0.27 ± 0.03 a	0.42 ± 0.05 b	nd	nd	nd	−0.4453	−0.3814
34.912	Indole	3.11 ± 0.23 d	1.18 ± 0.14 c	0.86 ± 0.05 b	0.60 ± 0.04 a	nd	nd	nd	−0.7501	−0.7868
35.511	1H-Indole, 3-methyl-	1.76 ± 0.28	nd	nd	nd	nd	nd	nd	−0.5316	−0.6052

Data expressed as mean values (*n* = 3). RT—retention time; nd—not determined; r—Pearson correlation coefficient; OA—overall acceptability; VC—volatile compound; ‘H’—emotion ‘happy’ fixed by FaceReader. a–f. Mean values within a column, between all the tested bread samples, with different letters are significantly different (*p* ≤ 0.05). The control bread was produced without wheat bran; WB5%, WB10%, WB15%—bread produced with 5%, 10%, and 15%, respectively, untreated wheat bran; WB ex-f-5%, WB ex-f-10%, WB ex-f-15%—bread produced with 5%, 10%, and 15%, respectively, extruded fermented wheat bran.

## Data Availability

The data are available from the corresponding author upon reasonable request.

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
