# Peer review of "The Contribution of Extruded and Fermented Wheat Bran to the Quality Parameters of Wheat Bread, Including the Profile of Volatile Compounds and Their Relationship with Emotions Induced for Consumers"

_foods, 2021, doi:10.3390/foods10102501_

Round 1
Reviewer 1 Report
In my opinion, the manuscript is very well planned, well-written and after some minor changes. It fits well with the scope of the Journal.
I would suggest improving Table 3 to make it easier to read. Additionally, consider highlighting the most important values.
Author Response
Reviewer 1: In my opinion, the manuscript is very well planned, well-written, and after some minor changes. It fits well with the scope of the Journal.
I would suggest improving Table 3 to make it easier to read. Additionally, consider highlighting the most important values.
Authors response: Authors are thankful for the comment, we would like to explain that according to Foods journal rules, we can not highlight values in Tables, because during the proofing service they asking to give all the values without bold, Italic, etc. highlighting.

Reviewer 2 Report
It is of crucial importance to improve the sensory quality of wheat bread with high fibre content. The use of pre-treatment technologies, such as extrusion and fermentation of wheat bran, revealed to be a promising approach. Considering the commercial, sensory and nutrition advantages of sourdough incorporation in bread formulations, this extruded and fermented bran ingredient could be considered an alternative sourdough. The article contains new information, the aim of the work is clearly established, the experimental well conducted, discussion and conclusions are well documented and scientifically coherent, despite some concerns:
Introduction
Lines 42-44: Authors say: “However, despite most of the functional compounds of the wheat grain being located in the outermost tissues [1], the other layers of wheat grain have not, till now, been used effectively enough”. Instead of “the other layers”, I suggest “these layers”.
Lines 56-58: Authors say: “Also, an extrusion process can be applied for wheat bran pre- treatment, as during this process there are microbial, structural, and physicochemical changes to the substrate [6]”. I think you should remove “microbial” since it is related to fermentation in vitro studies [6] and not to extrusion process.
Materials and methods
Bread preparation, Lines 91-93: In bran formulations, was wheat flour replaced by wheat bran? Did the Authors fixed the water content for all wheat bread formulations at 56%? How it was determined, by using farinographic tests? Is this value referred to 14% water basis? This should be detailed.
In my opinion this point is of crucial importance because the type and level of wheat bran will influence the water absorption (WA). In bread industry the WA adjustment depending on bread formulation is a current and important practice since it influences dough behaviour and bread quality parameters. This doesn´t comprise your study and findings because WA can be fixed for all formulations, but do you expect different results with WA adjustment?
Evaluation of bread quality parameters, Lines 113-118: Which probe was used for TPA?
Results and discussion
3.1 Section and Table 1: In Material and methods section, Lines 116-118, Authors say: “The resulting peak force of compression was reported as crumb firmness. Three replicates from three different sets of baking were analysed and averaged”. However, this section is related to dough (?!)… In table 2: n=5, results for hardness are related to dough, and unit is mJ (energy instead of Force - N in SI units). In addition, “a–e Mean values within a row with different letters are significantly different (p ≤ 0.05)”, row should be replaced by column, and I don´t understand the criterium used to a-e letters (usually, the higher value corresponds to “a”).
Lines 159-172 and 238-251: In my opinion it would be more useful to calculate and comment total colour difference values (ΔE*) between each sample and Control sample.
Lines 181-184: Authors say: “A first characteristic of wheat bran which should be pointed out is its water absorption, which is different to that of endosperm flour [2]. However, after extrusion, as well as after fermentation, wheat bran characteristics change, and in this study, the hardness of dough prepared with extruded/fermented wheat bran was the same as that of the control dough”. In my opinion, something should be said about the WA adjustment…
3.2 Section, Table 2 and Figure 2: I think row should be replaced by column in Table 2 caption, “a–e Mean values within a row with different letters are significantly different (p ≤ 0.05)”. Hardness has not unit in Figure 2, attention to the numbers (e.g., 2.2 instead of 2,2) and nothing is referred about ANOVA letters.
3.3 Section and Table 3: Row should be replaced by column in Table 3 caption.
Table 4: “a–f Mean values within a column, between all the tested bread samples, with different letters are significantly different (p ≤ 0.05)”. I think column should be replaced by row.
Author Response
Reviewer 2. It is of crucial importance to improve the sensory quality of wheat bread with high fibre content. The use of pre-treatment technologies, such as extrusion and fermentation of wheat bran, revealed to be a promising approach. Considering the commercial, sensory, and nutrition advantages of sourdough incorporation in bread formulations, this extruded and fermented bran ingredient could be considered an alternative sourdough. The article contains new information, the aim of the work is clearly established, the experimental well-conducted, discussion and conclusions are well documented and scientifically coherent, despite some concerns.
Authors response: Authors are thankful for valuable comments.
Reviewer 2. Introduction Lines 42-44: Authors say: “However, despite most of the functional compounds of the wheat grain being located in the outermost tissues [1], the other layers of wheat grain have not, till now, been used effectively enough”. Instead of “the other layers”, I suggest “these layers”.
Authors response: corrected.
Reviewer 2. Lines 56-58: Authors say: “Also, an extrusion process can be applied for wheat bran pre-treatment, as during this process there are microbial, structural, and physicochemical changes to the substrate [6]”. I think you should remove “microbial” since it is related to fermentation in vitro studies [6] and not to extrusion process.
Authors response: corrected.
Reviewer 2. Materials and methods. Bread preparation, Lines 91-93: In bran formulations, was wheat flour replaced by wheat bran? Did the Authors fixed the water content for all wheat bread formulations at 56%? How it was determined, by using farinographic tests? Is this value referred to 14% water basis? This should be detailed. In my opinion, this point is of crucial importance because the type and level of wheat bran will influence the water absorption (WA). In bread industry, the WA adjustment depending on bread formulation is a current and important practice since it influences dough behaviour and bread quality parameters. This doesn´t comprise your study and findings because WA can be fixed for all formulations, but do you expect different results with WA adjustment?
Authors response: Authors are thankful for valuable comment, an explanation was included:
The wheat bread recipe consisted of 1 kg of wheat flour, 1.5% salt, 2% fresh compressed yeast, and 56% water (control bread). Control samples were made without the addition of wheat bran. Different quantities of the extruded / fermented wheat bran were added to the wheat flour (wheat flour 100% plus WB ex-f-5%, WB ex-f-10%, WB ex-f-15%. The water content was calculated as 56% of the total flour and wheat bran mass.
Reviewer 2. Evaluation of bread quality parameters, Lines 113-118: Which probe was used for TPA?
Authors response: Authors are thankful for comment, modifications of explanation are given:
Bread slices of 2 cm thickness were compressed to 10% of their original height at a crosshead speed of 10 mm s−1.
Reviewer 2. Results and discussion 3.1 Section and Table 1: In Material and methods section, Lines 116-118, Authors say: “The resulting peak force of compression was reported as crumb firmness. Three replicates from three different sets of baking were analysed and averaged”. However, this section is related to dough (?!)… In table 2: n=5, results for hardness are related to dough, and unit is mJ (energy instead of Force - N in SI units). In addition, “a–e Mean values within a row with different letters are significantly different (p ≤ 0.05)”, row should be replaced by column, and I don´t understand the criterium used to a-e letters (usually, the higher value corresponds to “a”).
Authors response: Authors are thankful for comment, description was corrected:
Dough hardness was measured as the energy required for the sample deformation (CT3 Texture Analyzer, Brookfield, USA). 50 g of dough was placed ant the Petri dishes and compressed to 10% of their original height at a crosshead speed of 10 mm s−1. The resulting peak energy of compression was reported as dough hardness. Five replicates of dough were analysed and averaged.
Bread crumb firmness during storage in plastic packaging at room temperature (22 ± 2 °C) was determined as the energy required for the sample deformation (CT3 Texture Analyzer, Brookfield, USA). Bread slices of 2 cm thickness were compressed to 10% of their original height at a crosshead speed of 10 mm s−1. The resulting peak force of compression was reported as crumb firmness. Three replicates from three different sets of baking were analysed and averaged.
According to statistic, we would like to explain that comparison is between the row (different dough samples), not between the different parameters.
Reviewer 2. Lines 159-172 and 238-251: In my opinion it would be more useful to calculate and comment total colour difference values (ΔE*) between each sample and Control sample.
Authors response: We would like to explain that differences are described:
In comparison with control group dough, the lightness (L*) of the bread doughs prepared with untreated and extruded/fermented wheat bran was, on average, 11.8% and 8.1% lower, respectively. Comparing the redness and/or greenness (a* or −a*, respectively) of the different groups of dough with that of the control group, in all cases smaller differences were obtained for the doughs prepared with extruded/fermented wheat bran, doughs prepared with 5% extruded/fermented wheat bran showing 21.8 times lower greenness (−a*) that that of the control group. The highest redness (a*) was established for the dough samples prepared with 15% untreated wheat bran; it was, on average, 48.2% higher than that of the doughs prepared with the same amount of extruded/fermented wheat bran. The opposite tendency was found for dough yellowness (b*); b* coordinates of doughs prepared with untreated wheat bran were, on average, 4.1% lower than those of the control group. And by increasing the proportion of extruded/fermented wheat bran in the dough formulation, the b* coordinates of the dough showed a tendency to increase, the highest values being found for the doughs prepared with 15% extruded/fermented wheat bran (22.75 NBS).
Reviewer 2. Lines 181-184: Authors say: “A first characteristic of wheat bran which should be pointed out is its water absorption, which is different to that of endosperm flour [2]. However, after extrusion, as well as after fermentation, wheat bran characteristics change, and in this study, the hardness of dough prepared with extruded/fermented wheat bran was the same as that of the control dough”. In my opinion, something should be said about the WA adjustment…
Authors response: Authors are thankful for comment, an explanation was given:
Also, it should be mentioned that in this study, by increasing wheat bran content in the main bread formula, water content, also, was additionally added.
Reviewer 2. 3.2 Section, Table 2 and Figure 2: I think row should be replaced by column in Table 2 caption, “a–e Mean values within a row with different letters are significantly different (p ≤ 0.05)”. Hardness has not unit in Figure 2, attention to the numbers (e.g., 2.2 instead of 2,2) and nothing is referred about ANOVA letters.
Authors response: We would like to explain according to Table 2, that comparison is between the row (different bread samples), not between the different parameters.
Authors are thankful for comment, Figure 2 was corrected.
Reviewer 2. 3.3 Section and Table 3: Row should be replaced by column in Table 3 caption.
Authors response: We would like to explain according to Table 3, that comparison is between the row (different bread samples), not between the different parameters.
Reviewer 2. Table 4: “a–f Mean values within a column, between all the tested bread samples, with different letters are significantly different (p ≤ 0.05)”. I think column should be replaced by row.
Authors response: We would like to explain according to Table 4, that comparison is between the different bread samples, not between the different parameters.

Reviewer 3 Report
The manuscript of E. Bartkiene and co-authors describe the effect of two treatments (fermentation and extrusion) on wheat bran to transform it into a valuable ingredient for bread preparation. The overall quality and the scientific-sounding of the paper is high. From this point of view, it was a sincere pleasure to read and referring their paper. Unfortunately, in my opinion, minor aspects require revisions.
At first, the Authors do not describe the fermentation process and the final characteristics of treated bran. Similarly, the extrusion method is only briefly depicted. In this regard, it is clear that the Authors cite other previously published material, but this forces the reader to search continuously for information rendering it a little boring and confusing the reading. Therefore, I suggest introducing the omitted details in the manuscript, e.g. in the supplementary material. The necessity of details is even more evident in the valuation of the results of the analysis of the emotions induced by the bread tasting using FaceReader software. Indeed, this approach is unusual, and the reader needs a more profound and convincing description.
I find other confusing factors in figure 2. For example, I suppose that the different letters labelling the bars indicate a statistical difference of the values, but this should be explained in the caption. In addition, in this figure, the DS bars are not indicated, and for example, it is not easy to understand the similarity between control at 48 h of storage and WBD5% at 12 h. I suppose that could be a consequence of a significant variability of the results of the three values, but this must be described and discussed.
In addition, the formatting of table 3 is challenging to read.
Minor revisions are furthermore necessary to the wordy, especially in its first part, abstract that have some tricky reading parts, like the ones from lines 62 to 68.
Finally, it could be interesting to have some rheological data about the doughs to interpret the actual effect of the treated bran and the control on the gluten structure.
Attention must be given to the formal text presentation too: for example, the use of the bracket (in all the text) seem to be confused and even the Authors names list is not ordered finishing with "and".
Author Response
Reviewer 3. The manuscript of E. Bartkiene and co-authors describe the effect of two treatments (fermentation and extrusion) on wheat bran to transform it into a valuable ingredient for bread preparation. The overall quality and the scientific-sounding of the paper is high. From this point of view, it was a sincere pleasure to read and referring their paper. Unfortunately, in my opinion, minor aspects require revisions.
Authors response: Authors are thankful for valuable comments.Reviewer 3. At first, the Authors do not describe the fermentation process and the final characteristics of treated bran. Similarly, the extrusion method is only briefly depicted. In this regard, it is clear that the Authors cite other previously published material, but this forces the reader to search continuously for information rendering it a little boring and confusing the reading. Therefore, I suggest introducing the omitted details in the manuscript, e.g. in the supplementary material.
Authors response: Supplementary material 1 was included.
Reviewer 3. The necessity of details is even more evident in the valuation of the results of the analysis of the emotions induced by the bread tasting using FaceReader software. Indeed, this approach is unusual, and the reader needs a more profound and convincing description.
Authors response: Supplementary material 2 was included.Reviewer 3. I find other confusing factors in figure 2. For example, I suppose that the different letters labelling the bars indicate a statistical difference of the values, but this should be explained in the caption. In addition, in this figure, the DS bars are not indicated, and for example, it is not easy to understand the similarity between control at 48 h of storage and WBD5% at 12 h. I suppose that could be a consequence of a significant variability of the results of the three values, but this must be described and discussed.
Authors response: Authors are thankful for comment, Figure 2 was corrected.Reviewer 3. In addition, the formatting of table 3 is challenging to read.
Authors response: We would like to explain that this is the best way to show results, because results broad variety and it is not possible to give this results in Figures, as well as, according to Foods journal rules, it is not possible to highlight them (Bold or Italic is not allow).
Reviewer 3. Minor revisions are furthermore necessary to the wordy, especially in its first part, abstract that have some tricky reading parts, like the ones from lines 62 to 68.
Authors response: corrected.
Reviewer 3. Finally, it could be interesting to have some rheological data about the doughs to interpret the actual effect of the treated bran and the control on the gluten structure.
Authors response: Authors agree with Reviewers’ comment, however, this analysis was not included in this experiment. However, we are thankful for idea, which we will use in the nearest future.
Reviewer 3. Attention must be given to the formal text presentation too: for example, the use of the bracket (in all the text) seem to be confused and even the Authors names list is not ordered finishing with "and".
Authors response: Authors are thankful for comment, corrections were made.
